# PARAMETER-EFFICIENT REINFORCEMENT LEARNING USING PREFIX OPTIMIZATION

**Itamar Rocha Filho**
Harvard University

**Rosie Zhao**
Harvard University
Kempner Institute

**Sham M. Kakade**
Harvard University
Kempner Institute

**Eran Malach**[†]
Harvard University
Kempner Institute

**Samy Jelassi**[†]
Harvard University

## ABSTRACT

Reinforcement Learning with Verifiable Rewards (RLVR) is a leading approach for tuning language models on mathematical reasoning tasks. However, it remains unclear whether RLVR's gains stem from genuine reasoning improvements or simply from steering the model toward answer formats that already appear in the reference distribution. Inspired by recent evidence (Zhao et al., 2025; Yue et al., 2025), we study this question by optimizing only the first $k$ tokens (e.g. $k = 32$) of each solution, generating the remainder of the response from the reference model. We study two methods for prefix optimization, using a naive algorithm that clusters prefixes and selects the best prefix (Prefix Clustering), and a method that optimizes the prefix by finetuning a lightweight adapter model with RL (Prefix-RL). We show that tuning only the first $k$ tokens can significantly improve the accuracy on math, suggesting that at least some of the gains from RL are due to upweighting a preferable solution strategy. Our results suggest that simple prefix optimization methods can provide an efficient alternative to RL, delivering substantial improvements across different models and benchmarks for a tiny fraction of the compute required for standard RL, and that these gains are robust across prefix lengths and random seeds.

## 1 INTRODUCTION

Reinforcement Learning (RL) based finetuning is used for improving language models performance on mathematical reasoning (Jaech et al., 2024; Guo et al., 2025; Shao et al., 2024; Team et al., 2025), coding (Austin et al., 2021; Gehring et al., 2024; Luo et al., 2025) and other domains (Ouyang et al., 2022; Lee et al., 2024; Bai et al., 2022; Gurung & Lapata, 2025; Su et al., 2025). A recent work studying the behavior of RL on mathematical reasoning has observed that the improvement due to RL can be attributed, at least partially, to a process of upweighting beneficial behaviors or strategies that are learned in the pretraining phase (Zhao et al., 2025). In particular, the authors show that RL causes the models to concentrate their output distribution on particular generation formats that achieve higher relative accuracy. These results are demonstrated for models pretrained "from scratch" on different mixtures of datasets that contain a variety of strategies for solving math problems, such as using different styles of code and text.

We begin this work by investigating to what extent the gains from RL can be attributed to upweighting useful strategies that are already present in the reference model. To test this, we evaluate different approaches for improving the model's performance by optimizing the *prefix* of the response. In other words, our goal is to change only the first $k$ tokens generated by the model after prompted with the input question, for some small $k$. Indeed, note that the first $k$ tokens typically reveal the solution strategy and format (e.g., starting with "## Step 1: To..." indicates a step-by-step solution).

---

[†]Equal advising.

Code available at `https://github.com/ItamarRocha/Prefix-RL`. This repository provides a reference implementation.

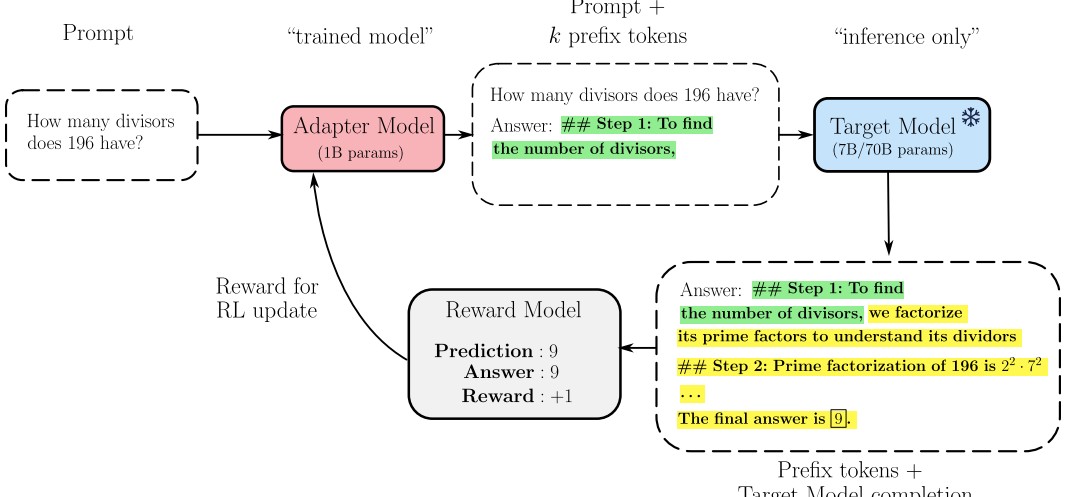

Figure 1: An illustration of the Prefix-RL method. A small *adapter* model receives the input question and generates the first $k$ tokens (prefix) in the response. The large *target* model is used in "inference-mode" to complete the solution. We then compute the reward based on the final answer generated by the model, and apply standard RL procedure (PPO) to update the *adapter* model (keeping the target model frozen).

Therefore, optimizing only the prefix captures the improvement due to upweighting good strategies. Since in our setting the majority of the tokens come from the reference model, RL cannot be used to improve genuine reasoning skills.

We test two methods for prefix optimization. First, we try a very naive approach that selects a *fixed* prefix (i.e., a prefix that does not depend on the input) and uses it as the beginning of the response for all input prompts. To do this, we perform *prefix clustering*: we cluster prefixes of 16 tokens generated by the reference model using a standard clustering algorithm into $5 - 6$ clusters, then we choose the single *fixed* prefix that maximizes performance on the MATH train set. Surprisingly, this extremely simple method that uses *the same* "optimized" prefix for all input prompts already achieves significant boosts in performance for certain models in the Llama family, as these models seem to improve simply by starting the response with a prefix that indicates step-by-step reasoning. However, this method does not improve performance on Qwen models, as the "preferred" prefixes for these models usually relate to the input more strongly.

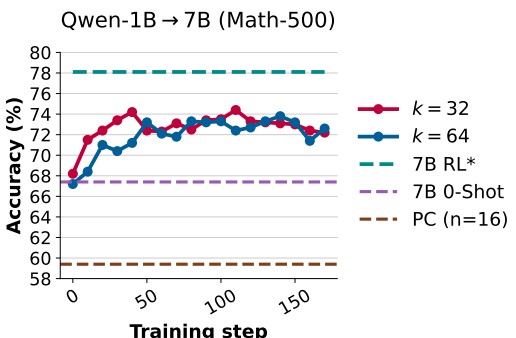

Figure 2: We finetune Qwen-1B using RL to generate the first $k$ tokens in the answer, with a frozen (inference-only) Qwen-7B model that completes the solution. *The accuracy of running RL on the full 7B model is taken from SimpleRL (Zeng et al., 2025), and we treat it as a skyline for our method.

As a next step, we finetune the different models using RL to generate only the first $k$ tokens (prefix) in an answer, and let the reference (pre-RL) model complete the generation. This method, which we call **Prefix-RL**, uses a small ($\approx 1B$ parameter) *adapter* model for generating a prefix that guides the generation of a much larger *target model*. We then use RL to optimize the small *adapter* model, while requiring only inference access to the larger *target model* (see Figure 1).

Interestingly, in this case our experiments show that significant improvements can be gained with many models by optimizing only a short prefix of tokens using RL (see Appendix Table 4). We note that due to limited computational resources, we were not able to perform complete RL finetuning

Table 1: Performance of Prefix-RL and Prefix-Clustering (PC) on different choices of reference models and math benchmarks. For all the Prefix-RL experiments, we use a 1B-parameter model from the same family as the adapter model for generating a prefix of $k$ tokens to the target model. We compare the model's performance before RL against the benchmarks, and for Qwen-7B, we also add LoRA and Prefix-Tuning baselines. Here, we report the performance of the best checkpoint. Regarding Prefix-Clustering, for each reference model, we sample $k=16$-token prefix candidates on MATH-train, cluster them with $k$-means (with $k$ being chosen via the elbow method (Thorndike, 1953)), evaluate, and fix the best single prefix for all test evaluations. Then, we let the reference model complete this prefix and report the accuracy over different math benchmarks.

| Target Model | Math-500 | AIME | AMC23 | Minerva |
|---|---|---|---|---|
| Qwen-7B | 67.4 | 23.0 | 40.3 | 19.1 |
| +Prefix-RL ($k=32$) | 74.4 (**+7.0**) | 25.8 (**+2.8**) | 50.0 (**+9.7**) | 22.4 (**+3.3**) |
| +Prefix-RL ($k=64$) | 73.8 (**+6.4**) | 23.3 (**+0.2**) | 52.2 (**+11.9**) | 22.1 (**+2.9**) |
| +Prefix Clustering ($k=16$) | 59.4 (**-8.0**) | 24.4 (**+1.4**) | 42.2 (**+1.9**) | 29.4 (**+10.3**) |
| +LoRA | 70.2 (**+2.8**) | 24.4 (**+1.4**) | 36.2 (**-4.0**) | 20.9 (**+1.8**) |
| +Prefix-Tuning | 45.4 (**-22.0**) | 4.93 (**-18.07**) | 20.62 (**-19.68**) | 13.97 (**-5.13**) |
| Llama-8B-Instruct | 48.4 | 17.3 | 23.4 | 14.0 |
| +Prefix-RL ($k=32$) | 50.8 (**+2.4**) | 20.7 (**+3.4**) | 27.5 (**+4.1**) | 19.9 (**+5.9**) |
| +Prefix-RL ($k=64$) | 51.0 (**+2.6**) | 20.7 (**+3.4**) | 26.9 (**+3.4**) | 20.2 (**+6.3**) |
| +Prefix Clustering ($k=16$) | 49.8 (**+1.4**) | 21.1 (**+3.9**) | 35.6 (**+13.1**) | 15.8 (**+1.8**) |
| Llama-8B-Instruct-FP8 | 43.6 | 17.3 | 31.2 | 14.0 |
| +Prefix-RL ($k=32$) | 50.4 (**+6.8**) | 19.1 (**+1.8**) | 26.9 (**-4.4**) | 18.4 (**+4.4**) |
| +Prefix-RL ($k=64$) | 49.8 (**+6.2**) | 20.7 (**+3.4**) | 25.6 (**-5.6**) | 18.8 (**+4.8**) |
| +Prefix Clustering ($k=16$) | 48.8 (**+5.2**) | 23.4 (**+6.1**) | 19.4 (**-11.9**) | 17.6 (**+3.7**) |
| Llama-70B-Instruct-FP8 | 62.0 | 32.8 | 45.0 | 29.0 |
| +Prefix-RL ($k=32$) | 67.8 (**+5.8**) | 49.1 (**+16.3**) | 46.2 (**+1.2**) | 34.6 (**+5.5**) |
| +Prefix-RL ($k=64$) | 68.4 (**+6.4**) | 48.2 (**+15.4**) | 47.5 (**+2.5**) | 32.4 (**+3.3**) |
| +Prefix Clustering ($k=16$) | 67.0 (**+5.0**) | 48.0 (**+15.2**) | 44.4 (**-0.6**) | 32.0 (**+2.9**) |
| Qwen-72B | 82.0 | 41.2 | 56.9 | 23.2 |
| +Prefix-RL ($k=32$) | 84.0 (**+2.0**) | 40.6 (**-0.5**) | 66.6 (**+9.7**) | 29.0 (**+5.9**) |
| +Prefix-RL ($k=64$) | 82.6 (**+0.6**) | 40.4 (**-0.8**) | 65.0 (**+8.1**) | 25.7 (**+2.6**) |
| +Prefix Clustering ($k=16$) | 71.8 (**-10.2**) | 43.0 (**+1.8**) | 58.4 (**+1.6**) | 30.9 (**+7.7**) |

on the models we test (of scale 7B and above). However, for Qwen-7B, we compare to the RL fine-tuning results reported by SimpleRL (Zeng et al., 2025), and see that prefix RL optimization improves performance from around 68% to 74%, with full RL finetuning offering further improvement to 78% (all on MATH-500). This shows that while RL still improves performance on later tokens, possibly through improvements in reasoning capabilities, a significant improvement is due to selecting a correct solution strategy and steering the models in the right direction.

The observation above, beyond being scientifically interesting, emphasizes the opportunity of improving models' performance by doing minimal optimization on the generation prefixes, instead of running full-finetuning with RL, which is often infeasible with a small computational budget. In particular, prefix optimization allows performance improvements with minimal compute, requires inference-only access to the optimized models, and results in no weight updates in the target model. While prefix optimization cannot directly replace full-model RL finetuning, it can provide a practical, accessible, and efficient alternative, enabling significant performance improvements even in resource-constrained scenarios. We believe that these findings can be leveraged for the development of a cheap and scalable alternative to RL, and leave a more thorough study of prefix optimization methods in different settings to future work.

## 1.1 RELATED WORK

**Controllable generation.** Early controllability methods steer a frozen LLM by adjusting per-token probabilities at decoding time. Plug-and-Play Language Models (PPLM) (Dathathri et al., 2019) perturb hidden states with gradient ascent, while GeDi (Krause et al., 2020) and DExperts (Liu et al., 2021) combine logits from small expert models with those of the base LM. These approaches achieve fine-grained control but must remain in the decoding loop for *every* token, increasing inference latency and hindering deployment on low-cost hardware. In contrast, Prefix-RL optimizes a *single* prefix before the backbone begins decoding, so the continuation proceeds at the native speed of the frozen model. See Liang *et al.* (Liang et al., 2024) for a comprehensive survey on controllable generation.

**Prompt-optimization.** Prompt-engineering and prompt-tuning techniques treat the *input prompt* as the object of optimization. RLPrompt (Deng et al., 2022) applies RL for optimizing discrete prompts, and Retroformer (Yao et al., 2023) trains a retrospective policy to rewrite prompts across dialogue turns. Prompts must be prepended to every user query, and their efficacy can be brittle across domains or small input perturbations. Prefix-RL instead optimizes a *prefix of the answer*, which is semi-structured and directly tailored to the task's solution space.

**Prefix-Tuning.** Li and Liang introduced Prefix-Tuning (Li & Liang, 2021), a lightweight alternative to full-model finetuning, where a small number of task-specific vectors (called the "prefix") are prepended to the model input, optimizing only these parameters while freezing the rest of the model. Unlike Prefix-Tuning, which optimizes continuous embeddings through supervised training, our Prefix-RL approach leverages RL with verifiable rewards to directly optimize the solution strategy in mathematical reasoning tasks, enabling effective steering of large models through minimal intervention.

**Adapter-based RL steering.** Inference-Time Policy Adapters (IPA) (Lu et al., 2023) learn a small network that rescales logits of a frozen LM at each step via RL. While IPA shares our frozen-backbone philosophy, it still requires step-wise intervention, so inference cost grows with sequence length. Prefix-RL intervenes only once, making it strictly cheaper at test time. Moreover, IPA is demonstrated on style and toxicity control, whereas we focus on *verifiable rewards for mathematical reasoning*, a setting where sparse, binary feedback is readily available from automated checkers.

**Parameter-efficient finetuning.** Low-rank and adapter-based methods such as LoRA (Hu et al., 2021), QLoRA (Dettmers et al., 2023), and adapter layers (Houlsby et al., 2019) reduce the number of *trainable* parameters by inserting small modules into a frozen backbone and only updating those modules. However, when used with RL, gradients must still be backpropagated through the entire target network, so the training FLOPs scale with the size of the backbone just as in full-model RL. In contrast, Prefix-RL performs RL updates only on a separate, small adapter that generates an answer prefix, while the large target runs in inference-only mode. This decouples the training cost from the target model size and enables RL steering even for very large or quantized targets.

## 2 THE EFFECTIVENESS OF PREFIX OPTIMIZATION WITH REINFORCEMENT LEARNING

We begin by empirically studying how Reinforcement Learning improves mathematical reasoning. In particular, we want to understand whether the improvements from RL are due to upweighting the probability of generating solution strategies that the reference (pre-RL) model can already generate, or whether RL improves reasoning capabilities (e.g. through improved arithmetic capability, better execution of proof steps, more accurate recall of mathematical facts, etc.). To test this, we will tune only the first $k$ tokens in the generation of the model, while generating the remainder of the solution from the reference model.

More formally, let $x_1, \ldots, x_n$ be a sequence of tokens encoding the input question, and let $f_{\text{ref}}$ be the reference model. Our goal will be to generate the first $k$ tokens of the output from some model $g_\theta$, i.e. $y_1, \ldots, y_k \sim g_\theta(x_1, \ldots, x_n)$. Then, we generate the rest of the response from the reference model, conditioned on the input and the prefix: $y_{k+1}, \ldots, y_m \sim f_{\text{ref}}(x_1, \ldots, x_n, y_1, \ldots, y_k)$. We then treat $y_1, \ldots, y_m$ as the complete solution and compute the reward $r(y_1, \ldots, y_m)$ based on this

solution (e.g., whether the answer to a mathematical question is correct). We consider two methods for sampling the prefix:

**Prefix Clustering.** Given some training dataset $D$ of inputs, for each input $x \sim D$ we sample an output from the reference model $y \sim f_{\text{ref}}(x)$. We cluster all the prefixes of length $k$ from the sampled outputs with $k$-means clustering (with $k$ being chosen via the elbow method (Thorndike, 1953)). Following this, we get $c$ prefixes, denoted $(y_1^{(1)}, \ldots, y_k^{(1)}), \ldots, (y_1^{(c)}, \ldots, y_k^{(c)})$. We then choose the prefix with the highest reward on the training set $D$:

$$i_{\max} = \arg\max_i \mathbb{E}_{x \sim D}\left[r\left(f_{\text{ref}}\left(x_1, \ldots, x_n, y_1^{(i)}, \ldots, y_k^{(i)}\right)\right)\right]$$

Then we use this *fixed*, input-independent prefix $y_1^{(i_{\max})}, \ldots, y_k^{(i_{\max})}$ for all examples, and compute the reward on the evaluation set. That is, we fix $g_\theta(x_1, \ldots, x_n) = y_1^{(i_{\max})}, \ldots, y_k^{(i_{\max})}$.

**Prefix-RL.** Here we use RL to finetune an *adapter* model $g_\theta$ to generate the first $k$ tokens of the output. When collecting rollouts during finetuning, we first generate $y_1, \ldots, y_k \sim g_\theta(x_1, \ldots, x_n)$ and then generate $y_{k+1}, \ldots, y_m \sim f_{\text{ref}}(x_1, \ldots, x_n, y_1, \ldots, y_k)$. We optimize the parameters of $g_\theta$ using PPO (Schulman et al., 2017) with respect to the rewards $r(y_1, \ldots, y_m)$, but keep $f_{\text{ref}}$ fixed for the entire procedure. In this section we set the adapter to have the same architecture and initialization as the reference model. Because $f_{\text{ref}}$ is frozen and only completes tokens $k+1{:}m$, any improvement can only come from changing *how* the sequence is started rather than from teaching the backbone new token-level skills. In other words, this setting isolates the extent to which RL gains can be explained by upweighting existing solution strategies in the pre-RL distribution. In the next section we will demonstrate how using small adapters (compared to the reference target model) can serve as a simple and effective alternative to standard RL. An illustration of this setup is shown in Figure 1.

The solution strategy style is typically determined by the first few tokens generated. Optimizing the prefix for the task we train on will allow the model to choose the best solution strategy for this task. However, since we are generating most of the tokens from the reference model, it is unlikely that our optimization can result in enhanced reasoning capabilities.

Figure 3 presents the accuracy achieved when optimizing only the first $k$ tokens of a sequence, using either *prefix clustering* or *prefix RL*, and compare it to RL applied to the entire sequence. While full-sequence RL ultimately yields the highest accuracy, both prefix optimization methods greatly improve performance compared to the reference model, demonstrating substantial gains with significantly less compute. This suggests that a notable portion of RL's benefit arises from guiding the model toward generating answers in more effective formats, rather than from enhancing reasoning capabilities.

Motivated by this insight, in the next section we will explore Prefix Clustering and Prefix-RL as efficient alternatives to standard RL. In particular, we will use Prefix-RL as a parameter-efficient alternative to RL, finetuning a small *adapter* model to generate a prefix for a larger *target* model. For both Prefix Clustering and Prefix-RL, we only require *inference access* to the target model, thus reducing the computational burden of training a large model.

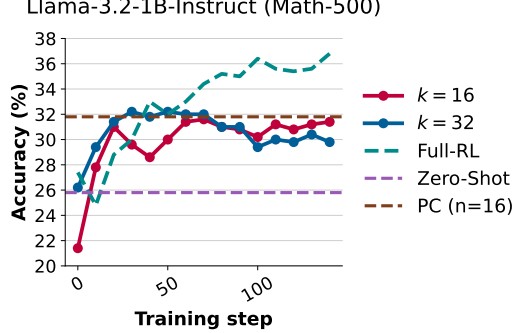

Figure 3: To test whether RL improves performance through upweighting of existing solution strategies, we RL-finetune a Llama-1B model to produce the first $k$ tokens in the response, using the reference (pre-RL) 1B model to complete the solution. We compare this against the performance of a standard RL procedure optimizing the full sequence of tokens, zero-shot, and a fixed Prefix-Clustering (PC) baseline.

Table 2: Comparison in terms of FLOPs between Standard RL finetuning and Prefix-RL with prefix of length $k$.

| Method | Training Compute | Inference Compute |
|---|---|---|
| Standard RL | $C_{\text{train}} N_t R T$ | $C_{\text{inf}} N_t R T$ |
| Prefix-RL $(k)$ | $C_{\text{train}} N_a R k$ | $C_{\text{inf}} R(N_t T + N_a k)$ |

$N_t$ = # Params. of target model, $N_a$ = # Params. of adapter model
$R$ = # Rollouts, $T$ = # Tokens per rollout.
$C_{\text{train}}$ = Train compute constant, $C_{\text{inf}}$ = Inference compute constant.

## 3 PREFIX-RL: SETTING AND RESULTS

We now turn to study the efficacy of Prefix Clustering and Prefix-RL on different choices of models and datasets. We start by describing the experimental setting and evaluation methods, introduce our results and discuss some analysis of the methods. We observe that while Prefix Clustering improves performance only for some models (namely, models from the Llama family), Prefix-RL provides improvements across most of the settings we try.

### 3.1 EXPERIMENTAL SETTING

**Models.** In our experiments, the adapter models are Llama-3.1-1B-Instruct (Grattafiori et al., 2024) and Qwen2.5-1.5B (Yang et al., 2024). The target models are Llama-3.1-8B-Instruct (Grattafiori et al., 2024), Llama-3.1-70B-Instruct-FP8 (NVIDIA, 2024), Qwen2.5-7B and Qwen2.5-72B (Yang et al., 2024). In all settings, we ensure that the adapter and target models belong to the same model family, as we observe (see section 4) that Prefix-RL does not perform as well when the two models are pretrained on different datasets. We also include quantized target models in our study—most notably Llama-3.1-70B-Instruct-FP8—highlighting a key strength of Prefix-RL: to our knowledge, this is the first demonstration of using RL to *steer* quantized FP8 models via a small learned adapter while keeping the quantized target weights frozen, showcasing the flexibility of our approach.

**Datasets.** We opt for the same dataset choices as in (Zeng et al., 2025). When the adapter model is Llama-3.1-1B-Instruct, the training dataset is the *training split* of MATH (Hendrycks et al., 2021), which contains 7,500 problems (the full MATH dataset has 12,500 problems across train and test). For Qwen, following Zeng et al. (2025), we build a larger RLVR dataset by taking all question–answer pairs from the MATH training and test splits with difficulty levels 3–5. We then remove any problems that appear in the MATH-500 evaluation set (Hendrycks et al., 2021; Lightman et al., 2023) to avoid contamination, resulting in 8,888 training examples for the Qwen2.5-1.5B adapter.

**Training.** We follow the OpenRLHF pipeline (Hu et al., 2024), using default coefficients of $\beta = 0.001$ for the KL divergence penalty and $\alpha = -0.001$ for the entropy bonus. Rollouts are generated using a sampling temperature of 0.7 with vLLM (Kwon et al., 2023). By default, we do not apply weight decay. The training and rollout batch sizes are both set to 512, and we sample 8 responses per prompt, resulting in 8 gradient updates per rollout step. We use the PPO algorithm (Schulman et al., 2017) and set the actor and critic learning rates as 5e-7 and 9e-6, respectively. By default, the maximum prompt length is set to 1600. For our standard RL experiments, the maximum response length is 2048. For Prefix-RL, the adapter is only allowed to emit the first $k$ tokens; unless otherwise stated we use $k \in \{32, 64\}$ for the main results in Table 1, and we additionally sweep $k \in \{1, 4, 8, 16, 32, 64\}$ for Qwen-7B in the robustness study in Appendix D. All experiments optimize a *verifiable* reward: after the target model finishes generation, we extract the final answer and compute a binary reward using `math_verify` (Kydlíček, 2024)—1 if the answer is verified correct, 0 otherwise. We store the model checkpoint every 10 steps for evaluation, and use 4 H100 GPUs for each experiment. We train all our models for 10 episodes which equates to 140 steps for the Llama adapter and 170 steps for the Qwen adapter.

We also train LoRA and supervised Prefix-Tuning baselines on Qwen-7B; see Appendices C.1 and C.2 for architecture and hyperparameters.

**Inference of the target model.** We serve the target models on separate nodes using vLLM. For small-sized target models as Llama-3.1-8B-Instruct and Qwen2.5-7B, the server requires 1 H100 GPU while larger models such as Llama-3.1-70B-Instruct-FP8 and Qwen2.5-72B, need 4 H100 GPUs. By default, the maximum generation length is set to 2048, the temperature to 0, $\texttt{top\_p}$ to 1 and only one completion is generated per prefix.

**Evaluation.** We evaluate our models on four widely used complex mathematical reasoning benchmarks: MATH-500 (Hendrycks et al., 2021; Lightman et al., 2023), AIME Problem Set 1983-2024 (Veeraboina, 2023), Minerva Math (Lewkowycz et al., 2022), OlympiadBench (He et al., 2024). For the adapter models, the maximum number of generated tokens is set to be the same as during finetuning and for the target model, the maximum number of generated tokens is set to 2048. We set the temperature to 0, $\texttt{top\_p}$ to 1, and only generate one pair (prefix, completion) per question. We use the $\texttt{math\_verify}$ package to extract the answers and verify their correctness. Lastly, we report the $\texttt{pass@1}$ performance in all our evaluations.

**Computational Benefits of Prefix-RL** Prefix-RL offers an alternative to standard RL that is substantially cheaper to run. Recall that Reinforcement Learning uses two distinct computational phases: inference—where we generate rollouts from the model to collect reward—and training—where we tune the parameters of the model using policy gradient. Table 2 summarizes the (approximate) cost of training and inference, which we discuss in Appendix B. Beyond a clear FLOPs advantage, Prefix-RL allows one to run RL finetuning with a smaller number of resources in practice. For instance, we are able to tune a 70B model using just 8 GPUs—4 for training the adapter and 4 for serving the target model. In contrast, standard RLHF implementations (Hu et al., 2024; Volcengine, 2024) typically require 32 GPUs for the same setup, representing a $4\times$ reduction in GPU usage. Moreover, we believe further efficiency gains are possible through improved allocation of compute between training and inference, as well as by sharing inference across processes—suggesting that the practical savings of Prefix-RL may be even greater.

## 3.2 MAIN RESULTS

We observe that Prefix Clustering achieves mixed performance: it improves performance (quite significantly) in some settings, but in some cases it results in severe degradation in performance. Prefix-RL, on the other hand, results in more consistent behavior. While the magnitude of the gains varies depending on the target model and on the specific task, Prefix-RL consistently yields performance improvements across the range of model scales and mathematical reasoning benchmarks that we tested.

**Performance gains with Prefix Clustering.** Prefix Clustering yields mixed results (Table 1). For the Llama family, a fixed $k{=}16$ prefix delivers sizable gains on several benchmarks, e.g., +13.1 on AMC23 for Llama-8B-Instruct and +15.2 on AIME for Llama-70B-Instruct-FP8, suggesting that these models benefit substantially from being steered into a particular solution style (typically an explicit step-by-step plan). In contrast, Prefix Clustering performs poorly on Qwen models (e.g., $-8.0$ on MATH-500 for Qwen-7B), indicating that Qwen's preferred openings are more input-dependent. Overall, PC is a compelling, training-free baseline that surfaces the "format upweighting" effect, but its brittleness across families motivates learning *input-conditional* prefixes with Prefix-RL. Throughout the paper we therefore treat PC primarily as a *diagnostic* for whether any fixed opening can move accuracy at all, rather than as a practical inference-time method. Longer fixed prefixes (e.g., 32 or 64 tokens) quickly became brittle in preliminary experiments, especially on Qwen, where a global prefix often conflicts with the input question and hurts accuracy, so we keep PC at $k{=}16$ in all settings.

**Performance gains with Prefix-RL.** Despite being significantly more cost-efficient than standard RL fine-tuning, Prefix-RL recovers a substantial fraction of its performance gains. As shown in Figure 2, full RL finetuning yields a 10-point gain in accuracy (from 68% to 78%), while Prefix-RL, at a much lower training cost, achieves a 7-point improvement (from 68% to 75%).

Results from Table 1 show that the benefits of Prefix-RL extend to more challenging benchmarks; for instance, applying Prefix-RL to Qwen2.5-7B leads to a $+4.4\%$ in performance on Minerva Math. These gains are consistent across model families, with Llama-3.1-8B-Instruct also showing reliable improvement. Notably, performance gains persist at scale when using Llama-3.1-70B-Instruct-FP8 and Qwen-2.5-72B as the target model. While the improvement on Qwen-2.5-72B is smaller, we

hypothesize this is due to the model's already high baseline performance on MATH (82% prior to RL), and datasets containing more complex problems could yield further improvement even in the Prefix-RL training regime.

For completeness, we include full training-dynamics curves in Appendix Figure 5, which illustrate that Prefix-RL produces stable improvement throughout training across both $k = 32$ and $k = 64$ settings.

**Prefix-RL on quantized FP8 models.** In Table 1, the largest relative performance gains by Prefix-RL are observed with quantized FP8 target models, reaching up to a $+16.3\%$ increase in accuracy. This substantial improvement may be partially explained by the lower baseline performance of quantized model. For instance, the gap between Llama-3.1-8B-Instruct and its quantized counterpart is 5% on MATH-500 prior to finetuning (see Table 1). However, after applying Prefix-RL, the quantized model nearly closes this gap— maintaining only a 1% difference compared to the full-precision model.

**Robustness across random seeds.** To quantify variance in our most important configuration (Qwen2.5-1.5B $\rightarrow$ Qwen2.5-7B), we train Prefix-RL with four random seeds for $k \in \{32, 64\}$. Table 6 reports mean $\pm$ standard deviation over seeds, and Figure 6 plots the corresponding accuracy curves. Across all four benchmarks and both values of $k$, every seed outperforms the base model, and effect sizes (e.g., $+5.1$–$+10.6$ points on AMC23) are $2$–$25\times$ larger than the standard deviations. This indicates that, despite noisy per-step curves typical of RL, the gains from Prefix-RL are statistically robust to initialization.

**Sensitivity to prefix length.** We also study how performance depends on the prefix length $k$ for Qwen-7B. Using the same training setup as in Table 1, we train Prefix-RL with $k \in \{1, 4, 8, 16, 32, 64\}$ on MATH and evaluate the resulting checkpoints on MATH-500, AIME, AMC23, and Minerva. The results are summarized in Table 8 and Figure 9.

On MATH-500 (the training distribution), all values of $k$ improve over the 67.4% base accuracy, with scores clustered in a narrow band around 72–74%. However, the very short prefix $k{=}1$ behaves differently on the held-out benchmarks: it yields only modest gains or even *worse* performance on AIME ($-1.0$), AMC23 ($-0.9$), and Minerva ($-2.2$), even though those evaluations use the same checkpoints trained on MATH. In contrast, moderate prefix lengths $k \in \{4, 8, 16, 32, 64\}$ consistently improve all four benchmarks, with AMC23 gains ranging from $+1.3$ to $+7.5$ points and Minerva gains up to $+4.8$ points.

Together with the training-dynamics curves in Figure 8, which show stable critic/policy losses and smoothly increasing KL divergence for all $k$, this suggests that Prefix-RL is not overly sensitive to the exact choice of $k$ as long as the prefix is long enough to encode a solution strategy (roughly $k \geq 4$). In practice, selecting a coarse value such as $k{=}32$ works well across models and benchmarks, while extremely short prefixes ($k{=}1$) appear brittle, especially on out-of-distribution evaluations.

**OOD Non-Math RLVR.** To test whether these prefix policies transfer beyond math, we also evaluate the same Qwen-7B checkpoints (trained only on MATH) on two undergraduate physics benchmarks: the OCW Courses dataset (physics questions derived from MIT OpenCourseWare, released on HuggingFace) and UGPhysics (Xu et al., 2025). For each prefix length $k \in \{1, 4, 8, 16, 32, 64\}$, we keep the adapter fixed and only vary the number of hint tokens used at inference time. As shown in Table 7 and Figure 7, all prefix lengths improve accuracy on OCW Courses, and all $k$ except 64 improve UGPhysics over the zero-shot Qwen-7B baseline. The strongest gains again occur for moderate prefix lengths ($k \approx 8$–$32$), supporting the view that the learned prefixes encode a broadly useful solution strategy rather than overfitting to the MATH training distribution.

**Comparison to LoRA and supervised Prefix-Tuning.** Table 1 also reports parameter-efficient baselines that modify the Qwen2.5-7B target directly. A LoRA-based RL run yields a modest improvement on MATH-500 (+2.8 points) and AIME (+1.4) and a small gain on Minerva (+1.8), but it underperforms Prefix-RL and even hurts AMC23 (–4.0). Our Prefix-Tuning SFT baseline performs substantially worse: training a 32-token prefix on the same MATH-filtered data degrades MATH-500 by 22 points and reduces accuracy on the other benchmarks as well. Importantly, both baselines require backpropagating through the full 7B backbone, so their training cost is comparable to standard RL, whereas Prefix-RL achieves larger and more consistent gains while updating only a 1.5B adapter.

In summary, Prefix-RL enables scalable and resource-efficient reinforcement learning by shifting the optimization burden to a lightweight adapter, making it a viable solution for finetuning large models under strict compute constraints.

## 3.3 ANALYSIS

To better understand how Prefix-RL influences model behavior, we conducted a qualitative analysis of the generated prefixes. Specifically, we examined whether the learned prefixes promote more structured and effective solution strategies across a range of problem types.

As shown in Table 4, Prefix-RL consistently steers the model responses toward clearer, more structured, and solution-oriented reasoning. In contrast, the reference model often produces incomplete or vague responses lacking clear direction or strategy. After applying Prefix-RL, responses more frequently include explicit planning steps—for example, identifying relationships between logarithmic equations, analyzing properties of functions in calculus problems, or invoking geometric principles appropriately. These changes improve both the interpretability and correctness of the model's reasoning.

That being said, the degree of improvement varies; in the algebraic roots example, the refinement is relatively modest, primarily emphasizing solution simplification rather than introducing a fundamentally new strategy. This suggests that Prefix-RL may offer diminishing returns in settings where the base model already exhibits a reasonable degree of structure. Overall, these qualitative findings support the effectiveness of Prefix-RL in prompting large models to adopt clearer and more strategic problem-solving approaches, particularly in cases where the reference model's outputs are underspecified or unfocused.

## 4 DISCUSSION AND LIMITATIONS

In this work we studied how simple prefix optimization methods can achieve significant performance gains on verifiable mathematical reasoning problems. We discussed two methods: Prefix Clustering, which uses a simple clustering method for finding an optimal input-independent prefix, and Prefix-RL, which uses RL to finetune a small adapter model that generates the prefix for a large, frozen, target model. We showed that these prefix optimization methods, and in particular Prefix-RL, can significantly improve performance across different choices of models and benchmarks. Importantly, these methods are extremely efficient in terms of compute, reducing the cost of training to a negligible fraction of the cost of training the full model, shifting most of the computational load to running inference. This offers remarkable gains in practice, as the cost of inference is dropping quickly due to innovations in inference hardware (Chitty-Venkata et al., 2024), inference-optimized model architectures (Huang et al., 2024) and algorithms for improving inference-time efficiency and utilization (Leviathan et al., 2023). Reducing the amount of compute required for training, which is almost exclusively done on expensive GPUs or TPUs, offers significant gains in cost and energy.

Another clear advantage of prefix optimization methods over standard RL-based finetuning is that they can adapt the target model's behavior without changing its weights. It has been observed in different settings that updating the model's weights when finetuning on a particular task can cause *catastrophic forgetting*, harming performance on unrelated tasks (Luo et al., 2023; Kalajdzievski, 2024). More relevant to our context, different works show that RL-finetuning can cause a drop in performance due to "alignment tax" (Askell et al., 2021), causes language drift (Lee et al., 2019) or language mixing (Guo et al., 2025), which may degrade performance on natural language tasks. Prefix optimization avoids these failures by providing an adaptation method that does not change the target model, where different adapters trained for different tasks can be deployed without causing harmful interference.

Finally, we note that our methods can be used on top of closed-weights models like GPT or Claude, since we only require inference access via API. We believe that this offers a great promise for cheap user-specific RL optimization, but leave a thorough exploration of this setting to future work.

## 4.1 LIMITATIONS

While our results demonstrate promising improvements, we acknowledge that our method has some inherent limitations. We do not claim that Prefix-RL can achieve performance gains that are comparable to RL-based finetuning of the full model. Therefore, we view Prefix-RL as a way to trade-off performance for computational cost, providing a cheap and efficient way to boost performance on downstream tasks, even for very large models. Unfortunately, we are unable to perform side-by-side comparison between Prefix-RL and standard RL on large models, as performing full RL-finetuning of 70B parameter models is beyond our computational budget.

A key limitation of our study's scope is that we focus on single-pass generation under RLVR. We do not attempt to model multi-pass reflective solvers where the model may backtrack, revise, or branch mid-solution. For such systems, early tokens may still steer the initial plan, but later reflection steps could override or refine that plan in ways our setup does not capture.

Additionally, we find that Prefix-RL relies on both the adapter and target models belonging to the same model family, likely due to the importance of shared response patterns. While we have not extensively tested cross-family configurations (e.g., Qwen-1B adapter with Llama-70B target), preliminary observations suggest such mismatches lead to degraded performance. Fortunately, many large models have corresponding distilled variants that retain similar behavior (Muennighoff et al., 2025; Team, 2025; Guo et al., 2025), which may help mitigate this constraint.

## ACKNOWLEDGMENTS

Kempner Institute computing resources enabled this work. SK, SJ, and RZ acknowledge support from the National Science Foundation Grant under award #IIS 2229881. SJ acknowledges funding supported by the Center of Mathematical Sciences and Applications. This work has been made possible in part by a gift from the Chan Zuckerberg Initiative Foundation to establish the Kempner Institute for the Study of Natural and Artificial Intelligence. IRF is supported by a Fundação Estudar Fellowship. RZ is supported by a Simons Investigator Fellowship, NSF grant DMS-2134157, DARPA grant W911NF2010021, and DOE grant DE-SC0022199. RZ is supported by a Kempner Institute Graduate Research Fellowship.

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

# A  APPENDIX

Table 3: Comparison of model responses before and after RL training

| Question | Reference Model | After Prefix-RL (k=32) |
|---|---|---|
| Convert the point $(0,3)$ in rectangular coordinates to polar coordinates. Enter your answer in the form $(r, \theta)$, where $r > 0$ and $0 \leq \theta < 2\pi$. | ## Step 1: Recall the conversion formulas between rectangular and polar coordinates. The conversion from rectangular coordinates $(x, y)$ to polar coordinates $(r,$ | ## Step 1: To convert the point $(0,3)$ from rectangular coordinates to polar coordinates, we need to understand the relationship between these two |
| Define $p = \sum_{k=1}^{\infty} \frac{1}{k^2}$ and $q = \sum_{k=1}^{\infty} \frac{1}{k^3}$. Find a way to write $\sum_{j=1}^{\infty} \sum_{k=1}^{\infty} \frac{1}{(j+k)^3}$ in terms of $p$ and $q$. | We have that 

 `\begin{align*}` 
 `\sum_{j = 1}^\infty` 

 `\sum_{k = 1}^\in` | ## Step 1: To find a way to write the given double sum in terms of $p$ and $q$, we need to understand the |
| If $f(x) = \frac{3x-2}{x-2}$, what is the value of $f(-2) + f(-1) + f(0)$? Express your answer as a common fraction. | ## Step 1: Evaluate f(-2) To find the value of $f(-2)$, we substitute $x = -2$ into the | ## Step 1: To find the value of $f(-2) + f(-1) + f(0)$, we need to first |
| How many positive whole-number divisors does 196 have? | (The divisors are not necessarily distinct.) ## Step 1: Find the prime factorization of 196 To find the number of positive whole-number div | ## Step 1: To find the number of positive whole-number divisors of 196, we need to understand what makes up a divisor. ## |
| The results of a cross-country team's training run are graphed below. Which student has the greatest average speed? [asy] ... [/asy] | The average speed of each student is calculated by dividing the distance traveled by the time taken. The average speed of Evelyn is 4.5 km/h, the | ## Step 1: To find the average speed of each student, we need to calculate the total distance traveled and the total time taken for each student |

Table 4: Comparison of Qwen model responses before and after RL training

| Question | Reference Model | After Prefix-RL (k=32) |
|---|---|---|
| Each of the two Magellan telescopes has a diameter of $6.5$ m. In one configuration the effective focal length is $72$ m. Find the diameter of the image of a planet (in cm ) at this focus if the angular diameter of the planet at the time of the observation is $45''$. | (The angular diameter of the Sun is $0.5''$.) To solve this problem, we need to use the concept of angular | To find the diameter of the image of a planet at the focus of the Magellan telescope, we need to use the relationship between the angular diameter of the planet |
| A white dwarf star has an effective temperature, $T_e = 50,000$ degrees Kelvin, but its radius, $R_{\text{WD}}$, is comparable to that of the Earth. Take $R_{\text{WD}} = 10^4$ km $\left(10^7\text{ m or }10^9\text{ cm}\right)$. Compute the luminosity (power output) of the white dwarf. Treat the white dwarf as a blackbody radiator. Give your answer in units of ergs per second, to two significant figures. | N/A | To compute the luminosity of the white dwarf, we need to use the Stefan-Boltzmann law, which relates the power output of a blackbody radiator |
| Preamble: A prism is constructed from glass and has sides that form a right triangle with the other two angles equal to $45°$. The sides are $L, L$, and $H$, where $L$ is a leg and $H$ is the hypotenuse. A parallel light beam enters side $L$ normal to the surface, passes into the glass, and then strikes $H$ internally. The index of refraction of the glass is $n = 1.5$. Compute the critical angle for the light to be internally reflected at $H$. Give your answer in degrees to 3 significant figures. | To determine the critical angle for the light to be internally reflected at the hypotenuse $H$ of the prism, we need to analyze the path of the | To determine the critical angle for the light to be internally reflected at the hypotenuse $H$, we need to use Snell's Law and the geometry |
| A particular star has an absolute magnitude $M = -7$. If this star is observed in a galaxy that is at a distance of 3Mpc, what will its apparent magnitude be? | Assume that the distance modulus formula is given by $\mu = 5\log_{10}(d) - 5$, where $d$ is the | Assume that the distance modulus formula is given by $\mu = 5\log_{10}(d) - 5$, where $d$ is the |
| Find the gravitational acceleration due to the Sun at the location of the Earth's orbit (i.e., at a distance of 1AU ). Give your answer in meters per second squared, and express it to one significant figure. | (The mass of the Sun is $1.99 \times 10^{30}$ kg and the mass of the | To find the gravitational acceleration due to the Sun at the location of the Earth's orbit (i.e., at a distance of (1 AU |

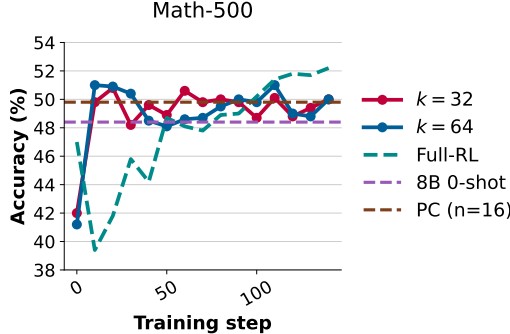

Figure 4: Training dynamics of Llama-3.1-8B-Instruct under Prefix-RL with $k \in 32, 64$ on MATH-500. We compare to full-sequence RL, zero-shot, and a fixed Prefix-Clustering baseline.

Table 5: Prefix-RL improves pass@1 on AIME 2024, AIME 2025, and OLYMPIADBENCH across model families. Numbers in parentheses are absolute $\Delta$ over the corresponding base model.

| Target Model | AIME2024 | AIME2025 | OlympiadBench |
|---|---|---|---|
| Qwen 7B | 7.5 | 3.8 | 14.7 |
| +Prefix-RL ($k = 32$) | 10.8 (**+3.3**) | 10.8 (**+7.1**) | 15.9 (**+1.2**) |
| +Prefix-RL ($k = 64$) | 9.2 (**+1.7**) | 10.0 (**+6.2**) | 15.4 (**+0.7**) |
| Llama 8B-Instruct | 10.0 | 0.0 | 5.2 |
| +Prefix-RL ($k = 32$) | 7.9 (**-2.1**) | 2.1 (**+2.1**) | 8.5 (**+3.3**) |
| +Prefix-RL ($k = 64$) | 8.8 (**-1.2**) | 2.1 (**+2.1**) | 7.7 (**+2.5**) |
| Llama 8B-FP8 | 6.7 | 0.0 | 5.6 |
| +Prefix-RL ($k = 32$) | 7.1 (**+0.4**) | 3.8 (**+3.8**) | 7.3 (**+1.6**) |
| +Prefix-RL ($k = 64$) | 7.9 (**+1.2**) | 1.7 (**+1.7**) | 7.7 (**+2.1**) |
| Llama 70B-Instruct-FP8 | 15.4 | 2.9 | 10.5 |
| +Prefix-RL ($k = 32$) | 20.0 (**+4.6**) | 3.8 (**+0.8**) | 12.6 (**+2.1**) |
| +Prefix-RL ($k = 64$) | 20.0 (**+4.6**) | 3.8 (**+0.8**) | 12.8 (**+2.2**) |
| Qwen 72B | 12.5 | 9.2 | 14.7 |
| +Prefix-RL ($k = 32$) | 12.5 (**+0.0**) | 14.6 (**+5.4**) | 19.4 (**+4.7**) |
| +Prefix-RL ($k = 64$) | 12.5 (**+0.0**) | 13.8 (**+4.6**) | 19.4 (**+4.7**) |

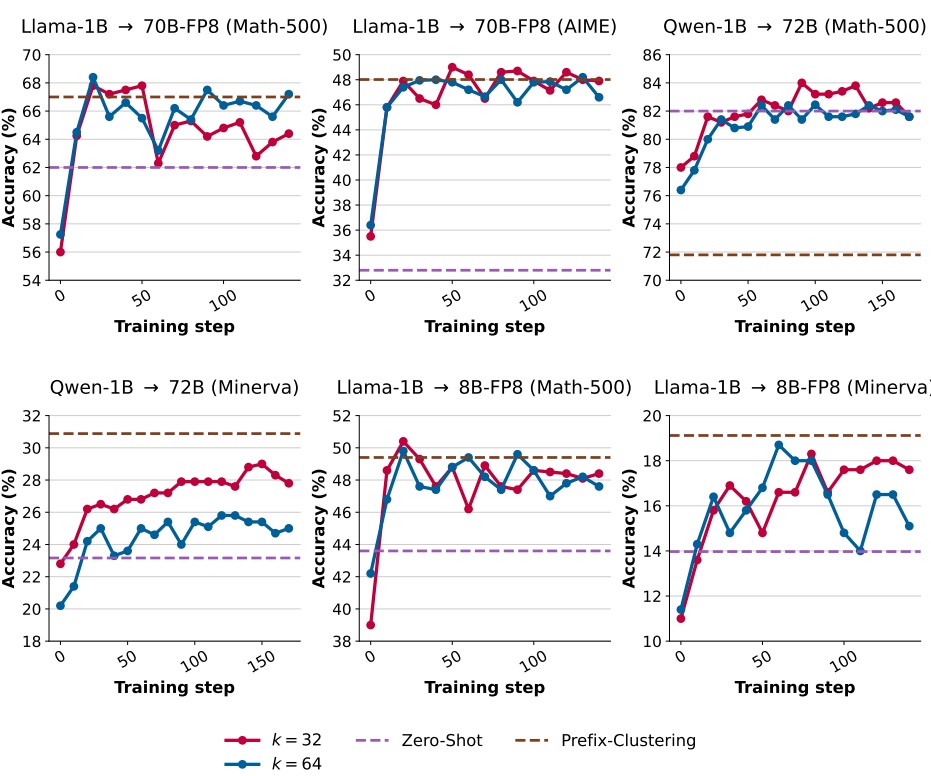

Figure 5: Training dynamics of Prefix-RL across models for prefix lengths $k = 32$ and $k = 64$.

## B   Computational Benefits of Prefix-RL

For the compute estimate, we use a common approximation for the FLOPs of transformer-based language models, where the FLOPs for training a model with $N$ parameters on $D$ tokens is $\approx 6ND$ (Kaplan et al., 2020; Hoffmann et al., 2022). Similarly, the inference cost for generating $D$ tokens with $N$-parameter model also grows approximately with $ND$, although with a different constant that may depend on the precision of the model and the hardware efficiency. Therefore, we can generally consider the training and inference cost to be $C_{\text{train}}ND$ and $C_{\text{inf}}ND$ respectively, where we use constants $C_{\text{train}}, C_{\text{inf}}$ that capture the efficiency of training and inference.

Now, let $N_t, N_a$ be the number of parameters for the target and adapter models respectively, and denote by $R$ the total number of rollouts and by $T$ the (average) number of tokens per rollout. Then, we get that the training compute for standard RL is $C_{\text{train}}N_t RT$ while the cost of training on prefixes of length $k$ using Prefix-RL is only $C_{\text{train}}N_a Rk$. Therefore, the number of training FLOPs is reduced by a factor $N_t T/(N_a k)$. By contrast, LoRA and other adapter-style approaches that attach modules directly to the target still require backpropagating through all $N_t$ backbone parameters at each step, so their training FLOPs match those of standard RL even though the number of *updated* parameters is smaller. For instance, when the adapter is Qwen2.5-1.5B and the target is Qwen2.5-72B, the FLOPs budget approximately $3,000$ times smaller than standard RL.

While training cost is reduced dramatically, Prefix-RL does involve a slight computational overhead in terms of inference FLOPs. Indeed, for every generation from the target model we need to first generate a prefix of $k$ tokens from the adapter, which adds an extra $N_a k$ FLOPs per rollout. However, note that these tokens are then used as an *input* to the target model, and can be processed in parallel instead of autoregressively. Therefore, overall inference time may actually be slightly *reduced* compared to standard RL, even though FLOPs are higher [This is similar to speculative decoding (Leviathan et al., 2023), where a small model generates a sequence of tokens that is validated by a larger model, where a (minor) increase in FLOPs enables a decrease in latency]. We note that a similar logic applies to inference calls to the model during serving (after the RL finetuning stage is over), where Prefix-RL can also provide a small improvement in inference latency.

## C   Extra Baselines

### C.1   LoRA RL Baseline

**LoRA RL baseline.** For Qwen2.5-7B we also train a parameter-efficient RL baseline using LoRA adapters attached directly to the target model. We follow the same RLVR pipeline as in our Prefix-RL runs (same reward, dataset, rollout configuration, and PPO hyperparameters), but instead of training a separate adapter we insert rank-64 LoRA modules with $\alpha = 32$ and dropout $0.0$ into the attention and feed-forward layers of Qwen2.5-7B and update only these parameters. Because gradients still flow through the full 7B backbone, the training FLOPs of this baseline scale with the target size $N_t$.

### C.2   Prefix-Tuning SFT Baseline

**Prefix-Tuning SFT baseline.** As a supervised counterpart to Prefix-RL, we train a Prefix-Tuning adapter on Qwen2.5-7B using the same MATH-filtered dataset. We follow the standard Prefix-Tuning setup (Li & Liang, 2021), with 32 virtual tokens, prefix projection enabled, and token/encoder dimensions matching the Qwen-7B hidden size. The model is trained with supervised fine-tuning (SFT) to predict full solutions, optimizing only the prefix parameters while keeping the backbone frozen. At evaluation time we attach the learned prefix adapter, decode deterministically, and score outputs with `math_verify` using the same prompts and evaluation pipeline as for Prefix-RL.

Table 6: Prefix-RL on Qwen-7B: mean $\pm$ standard deviation over 4 random seeds for $k \in \{32, 64\}$. All Prefix-RL configurations outperform the base model. The improvements are 2–25× larger than the corresponding standard deviations, indicating that our gains are robust to initialization.

| Target Model | Math-500 | AIME | AMC23 | Minerva |
|---|---|---|---|---|
| Qwen-7B | 67.4 | 23.0 | 40.3 | 19.1 |
| +Prefix-RL ($k = 32$) | $73.1 \pm 1.4$ (**+5.7**) | $24.5 \pm 0.4$ (**+1.5**) | $46.2 \pm 1.0$ (**+6.0**) | $23.3 \pm 2.6$ (**+4.2**) |
| +Prefix-RL ($k = 64$) | $72.5 \pm 0.2$ (**+5.1**) | $23.5 \pm 0.2$ (**+0.5**) | $50.9 \pm 2.4$ (**+10.6**) | $21.8 \pm 2.0$ (**+2.7**) |

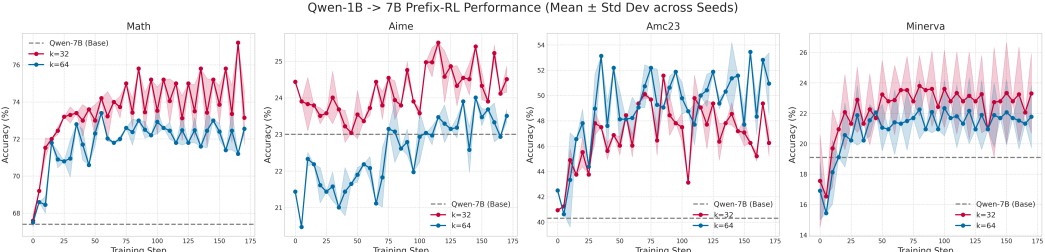

Figure 6: Prefix-RL training dynamics on Qwen-7B for $k \in \{32, 64\}$. Each panel shows accuracy vs. training step on a different math benchmark. Solid lines denote the mean over 4 random seeds; shaded regions represent the standard deviation.

# D  ADDITIONAL ROBUSTNESS EXPERIMENTS

## D.1  PREFIX-RL VARIATION ACROSS SEEDS

## D.2  PREFIX-RL K-SWEEP

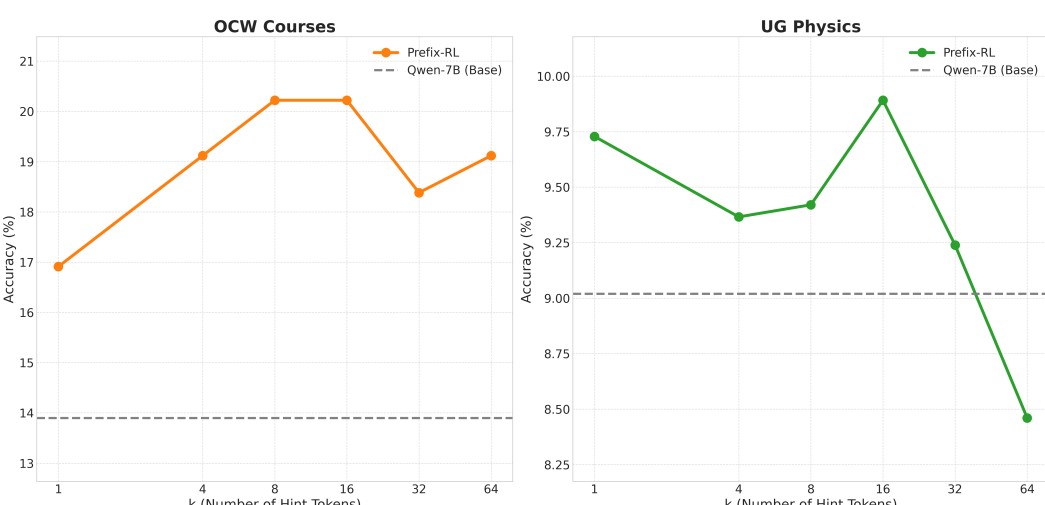

Figure 7: Prefix-RL transfer to physics benchmarks. We evaluate the same Qwen-7B checkpoints trained on MATH on two physics datasets—OCW Courses (Lewkowycz et al., 2022) (left) and UGPhysics (Xu et al., 2025) (right)—while varying the number of hint tokens $k$ used at inference time. The dashed line denotes the zero-shot Qwen-7B baseline. As in the math setting, moderate prefix lengths ($k \approx 8$–$32$) yield the largest gains, and all $k$ except $64$ improve over the base model on at least one dataset.

Table 7: Prefix-RL transfer to college-level physics benchmarks. We evaluate Qwen-7B on OCW Courses (Lewkowycz et al., 2022) and UGPhysics (Xu et al., 2025) using the same Prefix-RL checkpoints trained on MATH, varying the number of hint tokens $k$ at inference time. Parentheses show absolute improvement over the zero-shot Qwen-7B baseline.

| Target Model | OCW Courses | UG Physics |
|---|---|---|
| Qwen-7B | 13.9 | 9.0 |
| +Prefix-RL ($k = 1$) | 16.9 (**+3.0**) | 9.7 (**+0.7**) |
| +Prefix-RL ($k = 4$) | 19.1 (**+5.2**) | 9.4 (**+0.3**) |
| +Prefix-RL ($k = 8$) | 20.2 (**+6.3**) | 9.4 (**+0.4**) |
| +Prefix-RL ($k = 16$) | 20.2 (**+6.3**) | 9.9 (**+0.9**) |
| +Prefix-RL ($k = 32$) | 18.4 (**+4.5**) | 9.2 (**+0.2**) |
| +Prefix-RL ($k = 64$) | 19.1 (**+5.2**) | 8.5 (**-0.6**) |

Table 8: Effect of prefix length $k$ on Prefix-RL for Qwen-7B. Each row corresponds to a single Prefix-RL run with the given $k$. Numbers in parentheses are absolute changes in pass@1 compared to the base model.

| Target Model | Math-500 | AIME | AMC23 | Minerva |
|---|---|---|---|---|
| Qwen-7B | 67.4 | 23.0 | 40.3 | 19.1 |
| +Prefix-RL ($k = 1$) | 69.4 (**+2.0**) | 22.0 (**-1.0**) | 39.4 (**-0.9**) | 16.9 (**-2.2**) |
| +Prefix-RL ($k = 4$) | 72.8 (**+5.4**) | 22.5 (**-0.5**) | 44.7 (**+4.4**) | 20.2 (**+1.1**) |
| +Prefix-RL ($k = 8$) | 73.8 (**+6.4**) | 23.7 (**+0.7**) | 43.4 (**+3.1**) | 21.0 (**+1.9**) |
| +Prefix-RL ($k = 16$) | 73.0 (**+5.6**) | 23.2 (**+0.2**) | 41.6 (**+1.3**) | 23.9 (**+4.8**) |
| +Prefix-RL ($k = 32$) | 72.0 (**+4.6**) | 24.1 (**+1.1**) | 46.2 (**+5.9**) | 22.1 (**+3.0**) |
| +Prefix-RL ($k = 64$) | 72.8 (**+5.4**) | 23.7 (**+0.7**) | 47.8 (**+7.5**) | 19.9 (**+0.8**) |

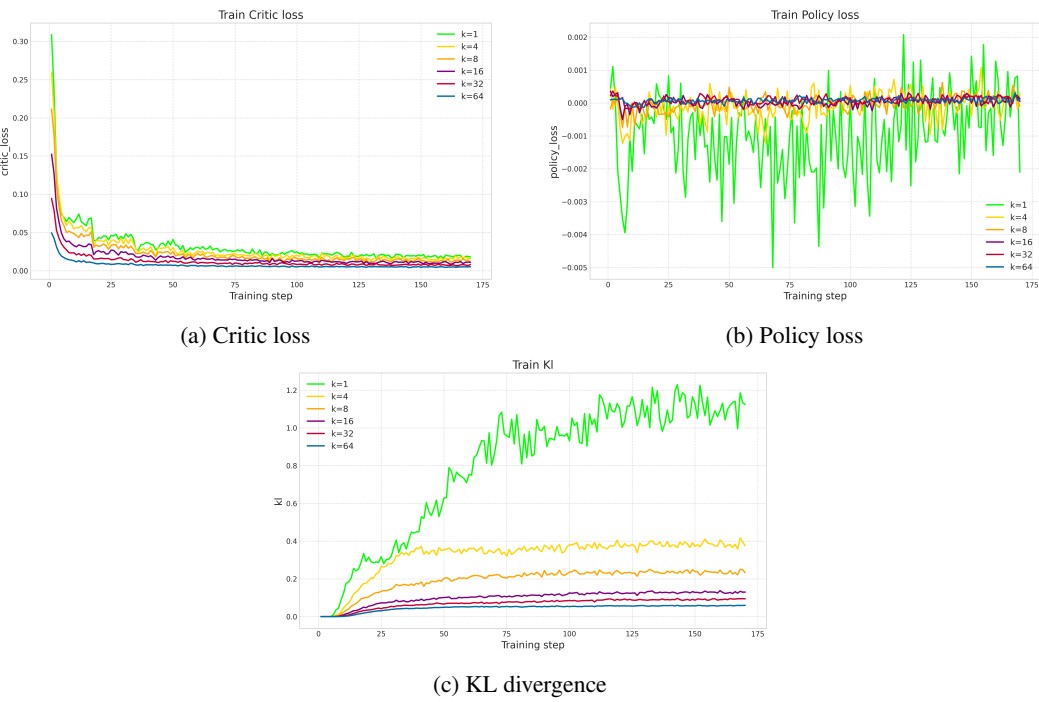

(a) Critic loss          (b) Policy loss

(c) KL divergence

Figure 8: Training dynamics of Prefix-RL on Qwen-7B for different prefix lengths $k \in \{1, 4, 8, 16, 32, 64\}$. Critic and policy losses quickly stabilize across all settings, while shorter prefixes (especially $k = 1$) induce a larger KL divergence from the reference policy.

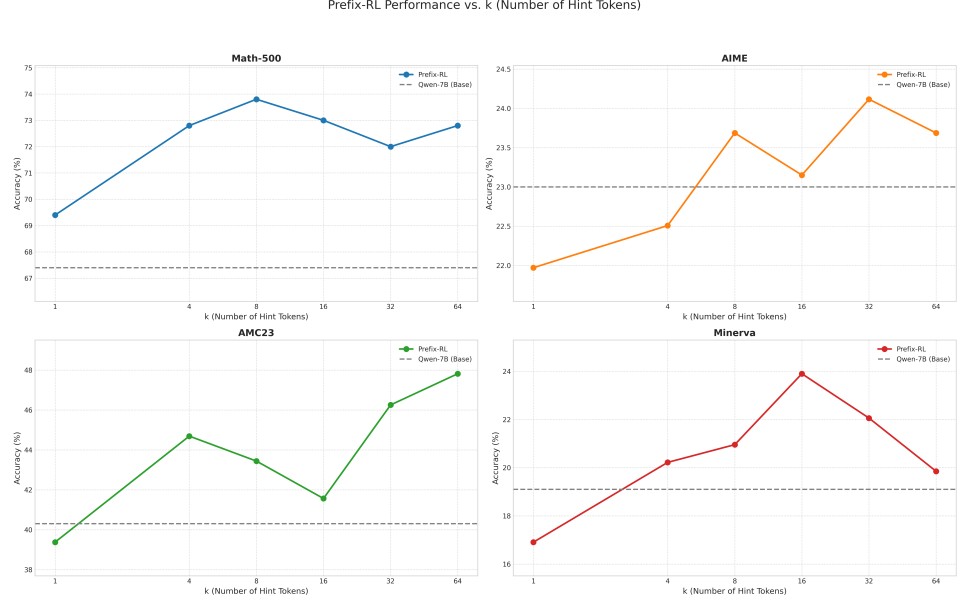

Figure 9: Prefix-RL performance vs. prefix length $k$ on Qwen-7B. Each panel shows pass@1 on a benchmark, using the same runs as in Table 8. Very short prefixes ($k = 1$) underperform on AIME, AMC23, and Minerva, while moderate lengths ($k \approx 4$–16) capture most of the gains.

## E USE OF LARGE LANGUAGE MODELS

LLMs were used solely for language editing and LaTeX assistance; all technical ideas, experiments, analyses, and conclusions are the authors' own, and all LLM outputs were reviewed and revised by the authors.

