# OpenReview forum: "Parameter-Efficient Reinforcement Learning using Prefix Optimization"
_ICLR.cc/2026/Conference — ICLR 2026 Poster_

### Official Review · Reviewer_UfLZ · 2025-10-31

**Soundness:** 2
**Presentation:** 3
**Contribution:** 2
**Rating:** 2
**Confidence:** 4

**Summary:**

This paper investigates parameter-efficient reinforcement learning (RL) for math reasoning by optimizing only the first k generated tokens (the “prefix”) and letting a frozen, larger target model complete the solution.

Two approaches are explored: (1) Prefix-Clustering, which selects a single fixed prefix by clustering candidate prefixes from the reference model and choosing the best on a training set; and (2) Prefix-RL, which RL-finetunes a small 1B adapter model to generate the first k tokens while the large target model remains frozen.

Using verifiable rewards (answer correctness), the authors show consistent gains on math benchmarks (e.g., MATH-500, AIME, Minerva) across Qwen and Llama families, including FP8-quantized ones, with significantly lower training compute than full-model RL. The empirical results suggest that a substantial share of RL gains arises from steering toward effective formats/strategies rather than improving token-by-token reasoning across the entire sequence.

**Strengths:**

- This paper introduces a compute-lean adapter-based RL setup where only k initial tokens are learned, separating strategy choice from long-horizon generation.
- The pipeline is well-illustrated (adapter emits prefix; target completes; reward computed on final answer).

**Weaknesses:**

1. The paper implicitly assumes early tokens determine the solution strategy that the model will keep following. This may not hold for reflective/iterative solvers (e.g., o3-like, DeepSeek-R1, Qwen-Thinking) that backtrack, revise, or branch mid-solution. The generality of prefix steering under multi-pass reflection remains untested.

2. Prefix-Clustering protocol seems train-set-dependent and of unclear inference value. The method traverses MATH-train to choose a single best fixed prefix, which may not be practically meaningful at inference time (and risks train-set over-selection).

3. To support the efficiency claim, add a direct baseline: “1B Prefix-RL + Large Target” vs. “Full RL on the Large Target” under matched or budget-normalized compute and matched data. Without this, the efficiency–performance trade curve is hard to judge.

4. Some figure narratives (e.g., the 1B self-completion plot analogous to Fig. 3) could better articulate what hypothesis each figure specifically tests (e.g., how much of full-RL gain is recovered by prefix control?)

5. The paper states this is the first demonstration of RL finetuning applied to quantized models yet the method does not RL-update the quantized target’s weights—only the small adapter is updated while the FP8 model is used for inference-only completion. As worded, this can be misleading.

**Questions:**

See weakness.

---

> ### Author Response · Authors · 2025-11-26
>
> Thank you for the thoughtful and constructive review. We appreciate your careful reading of the paper and the detailed suggestions. Below we address each of the main concerns and describe the corresponding changes in the revised manuscript.
>
> **Weakness 1: Assumption that early tokens determine solution strategy; multi‑pass reflective solvers**
>
> Our setup does not capture the behavior of highly reflective, multi‑pass solvers such as o3‑like models, DeepSeek‑R1, or Qwen‑Thinking. We have added explicit text in Section 4 stating that:
>
> * Our experiments deliberately isolate a single‑pass generation setting under RLVR.
> * The adapter only writes the first k tokens; the target then completes the response in one shot. We do not model later reflection, backtracking, or self‑critique loops.
>
> Thus, our conclusions are about strategy selection and formatting in single‑pass generation, not about full control of reflective pipelines.
>
> **Weakness 2: Train‑set dependence and limited practical value of Prefix‑Clustering**
>
> We agree that Prefix‑Clustering is train‑set‑dependent and not a practical inference‑time method. In the revised manuscript, we emphasize that we use Prefix‑Clustering purely as a diagnostic tool:
>
> * It answers the question “Can a single, fixed opening—chosen via performance on MATH‑train—move accuracy at all?”
> * A positive answer (especially for Llama) supports the idea that early tokens encode high‑value strategies.
>
> We explicitly state that Prefix‑RL is the method we expect practitioners to deploy, while Prefix‑Clustering is included only to probe the “format upweighting” phenomenon.
>
> **Weakness 3: Efficiency claims; need for budget‑normalized comparison**
>
> To substantiate the efficiency claims, we have:
>
> * **Table 2 and Appendix B**, which derive the training and inference FLOPs for standard RL vs Prefix‑RL. For Qwen‑1.5B → 72B, Prefix‑RL reduces the training FLOPs by roughly a factor of 3000, while inference cost is dominated by the target anyway.
> * A discussion in Section 3.1 explaining how we can tune a 70B FP8 target using just 8 GPUs (4 for the adapter, 4 for serving the target), whereas typical RLHF set‑ups require 32 GPUs under the same configuration, a 4× reduction in GPU count.
>
> **Weakness 4: Figure narratives and hypotheses**
>
> We have revised the figure captions and surrounding text to clearly articulate what each figure tests:
>
> * **Figure 1 / 3:** How much of full RL’s gain can be recovered by optimizing only the first k tokens? (≈70% on Qwen‑7B under ≈3000× lower training FLOPs.)
> * **Figure 4:** Training dynamics for different targets show that small prefixes (k ∈ {32,64}) consistently improve performance across benchmarks.
> * **Figure 5:** Comparison of full‑sequence RL, Prefix‑Clustering, and Prefix‑RL on Llama‑8B‑Instruct, showing that Prefix‑RL tracks much of the full RL improvement while being cheaper.
> * **Figures 6–8:** Robustness across random seeds and across prefix lengths k.
>
> We hope this clarified framing makes it easier to interpret the figures.
>
> **Weakness 5: Wording about RL finetuning on quantized FP8 models**
>
> You are correct that in our setup the FP8 target weights are never updated; only the small adapter is RL‑trained while the quantized target runs in inference‑only mode. We have revised the wording to:
> “this is the first demonstration of using RL to steer quantized FP8 models via a small learned adapter while keeping the quantized target weights frozen, showcasing the flexibility of our approach”
> This more accurately describes our contribution: we show that RL can effectively *steer* quantized FP8 models via a learned prefix, even though their internal weights are not modified.
>
> We hope that these clarifications resolve any misunderstandings about our contribution. If they do, we would kindly ask you to consider raising the overall score; if not, we would be very grateful for further guidance on any remaining concerns so that we can improve the work.

---

### Official Review · Reviewer_Yk1Z · 2025-10-31

**Soundness:** 3
**Presentation:** 4
**Contribution:** 4
**Rating:** 8
**Confidence:** 3

**Summary:**

This paper investigates whether the performance gains observed in RLVR for mathematical reasoning are due to genuine improvements in reasoning ability, or primarily from shifting the model toward high-accuracy solution strategies already present in the base distribution. To answer this, the authors propose prefix optimization: only the first k tokens of a generated solution are optimized, while the remainder is completed by a frozen reference model.

Two methods are evaluated:
1. Prefix Clustering — selects a fixed prefix via k-means clustering of sampled candidate prefixes and uses it for all inputs.
2. Prefix-RL — trains a small adapter using PPO to generate an input-conditional prefix conditioned on the question.

Despite modifying only a tiny fraction (first 16–64 tokens) of the sequence, both methods yield substantial accuracy improvements on math benchmarks such as MATH-500, AIME, AMC, Minerva, OlympiadBench, often recovering a large share of full RL gains. Prefix-RL is compute-efficient, works with quantized models, avoids catastrophic forgetting, and requires inference-only access to the main model. Improvements are most pronounced when the adapter and target share a model family.

Overall, the work argues that strategy selection and formatting, not deep reasoning skill, may explain a substantial portion of RL gains.

**Strengths:**

1. A simple enough method, especially Prefix Clustering, not only brings significant improvements on downstream tasks, but also unveils that the high-quality solution already learned in the pre-training distribution. It offers another profound insight into the origin.
2. Highly compute-efficient;
3. Could work with closed-weight models;

**Weaknesses:**

1. Lack of direct comparison to full RL at a large scale
2. Lack of comparison to other parameter-efficient RL methods.
3. Generalization beyond math remains uncertain;
4. Prefix clustering harms Qwen but helps Llama, suggesting architectural or data-distribution differences worth deeper investigation. I think more analysis on why Qwen behaves differently needs to be conducted.

**Questions:**

Same to Weakness.

---

> ### Author Response · Authors · 2025-11-27
>
> Thank you for the careful reading and very positive evaluation of the paper. We address each weakness below and highlight the corresponding changes in the revised version.
>
> ---
>
> ### **1\. Comparison to full RL at larger scale**
>
> We make the comparison to full RL explicit for Qwen‑2.5‑7B, the largest model for which public RLVR results exist. Figure 1 overlays Prefix‑RL with the SimpleRL 7B full‑RL curve: full RL improves MATH‑500 from 68→78%, while Prefix‑RL (1.5B adapter → 7B target, (k ∈ {32,64})) reaches ≈75%, recovering \~70% of the full‑RL gain at a tiny fraction of the training FLOPs.
>
> For 70B targets, running full RL is beyond our budget; we state this limitation clearly in Sec. 4.1. Instead, Table 2 and Appendix B derive the compute formulas, showing that Prefix‑RL reduces training FLOPs by a factor (≈ N\_t T /(N\_a k)), up to \~3000× for Qwen‑1.5B → 72B.
>
> ---
>
> ### **2\. Comparison to parameter‑efficient RL methods**
>
> In response to your question, we added explicit parameter‑efficient baselines on Qwen‑2.5‑7B (Table 1; details in App. C):
>
> * **LoRA‑RL baseline.** Rank‑64 LoRA (α=32, no dropout) is trained with the *same* RLVR setup as Prefix‑RL. It yields modest gains on MATH‑500 and AIME (67.4→70.2, 23.0→24.4) but **hurts** AMC23 (40.3→36.2) and gives only \+1.8 on Minerva.
>
> * **Supervised Prefix‑Tuning baseline.** A standard 32-token Prefix-Tuning adapter trained with SFT on the same auto-labeled data degrades performance substantially (e.g., MATH-500 67.4→45.4, AIME 23.0→4.9).
>
> Both baselines still backpropagate through the full 7B backbone, so their training FLOPs scale with (N\_t), like full RL. Prefix‑RL instead updates only a 1.5B adapter, never backpropagating through the target, yet achieves larger and more consistent gains across benchmarks.
>
> ---
>
> ### **3\. Generalization beyond math**
>
> We expanded our evaluation to probe generalization:
>
> * **Non‑math RLVR:** Table 7 and Fig. 7 evaluate the *same* Qwen‑7B Prefix‑RL checkpoints (trained only on MATH) on two undergraduate physics benchmarks, OCW Courses and UGPhysics. All (k) improve OCW, and all (k) except 64 improve UGPhysics; gains reach \+6.3 points on OCW for (k ∈ {8,16}).
>
> * **OOD math:** Table 5 shows results for AIME 2024, AIME 2025, and OlympiadBench. Prefix‑RL improves pass@1 in most settings, e.g., Qwen‑7B gains \+3.3 on AIME 2024 and \+7.1 on AIME 2025, while Llama‑70B‑FP8 gains \+4.6 on AIME 2024 and ≈+2 points on OlympiadBench.
>
> In Section 4, we now explicitly scope our claims to RLVR-style domains with verifiable rewards (e.g., math, physics, code) and present dialogue/generic RLHF as future work, rather than current claims.
>
> ---
>
> ### **4\. Why does Prefix‑Clustering help Llama but harm Qwen?**
>
> We agree that this behavior is intriguing and strengthened the analysis in Sec. 3.2 and Sec. 3.3.
>
> * **Empirical pattern.** Table 1 shows that a single fixed 16‑token prefix gives large gains for Llama (e.g., \+13.1 on AMC23, \+15.2 on AIME for Llama‑70B‑FP8), but *degrades* Qwen‑7B on MATH‑500 by −8.0.
>
> * **Qualitative inspection.** For Llama, the best PC prefixes are generic, task‑agnostic openings (“\#\# Step 1: …”) that simply push the model into a step‑by‑step format; this transfers well across problems. For Qwen, high‑reward prefixes tend to paraphrase key entities from the question or mention task‑specific quantities (examples in Tables 3–4), so a *global* 16‑token string often conflicts with new inputs and hurts accuracy.
>
> * **Design choice.** Motivated by this, we explicitly position Prefix‑Clustering as a *diagnostic* to test whether “any fixed opening can move accuracy at all,” keep it at (k=16), and rely on Prefix‑RL—input‑conditional prefixes—for practical use. Longer fixed prefixes (k ≥ 32\) were found empirically brittle, especially on Qwen, and we now say so in the text.
>
> ---
>
> We hope these new experiments and clarifications address your concerns and further support the paper’s claims. We would be happy to clarify any points that remain unclear.

---

### Official Review · Reviewer_y1zy · 2025-11-04

**Soundness:** 2
**Presentation:** 3
**Contribution:** 3
**Rating:** 4
**Confidence:** 4

**Summary:**

This paper proposes a method for performing RL with low computational resources. The approach optimizes a small model to generate the beginning portion of responses, after which a large model completes the remaining decoding. The authors experiment with Llama3.1 and Qwen2.5 series models on several mathematical reasoning tasks. Results show that prefix-RL can achieve most of the performance gains of standard RL using relatively little computational resources.

**Strengths:**

1. This paper proposes a method for RL under low computational resources that can achieve most of the performance gains with far less computational cost than conventional RL.
2. The proposed method does not require full access to the target model; it only needs inference access. Therefore, it is applicable not only to open-source models but also to closed-source models.
3. The paper designs a Prefix Clustering experiment to verify the importance of the beginning portion of the response for performance gains, and further proposes the main method of this work, Prefix-RL.

**Weaknesses:**

1. The method in this paper is limited by the need to use models from the same family, which may cause it to perform poorly or even fail in settings involving closed-source models.
2. Experiments in this paper were conducted only on mathematical reasoning tasks, so the method's applicability to other RLVR tasks remains unknown.

**Questions:**

1. The upper-right subplot of Figure 4 shows an anomalous behavior of Prefix Clustering on the Qwen model; in L375–L376 the paper explains this as "Qwen’s preferred openings are more input-dependent." Could a clearer example be provided to substantiate this point?
2. If the target model were used directly as the adapter model, what kind of performance could be expected? This approach seems to potentially eliminate the need for an additional model and avoid the restriction that the method requires a smaller model within the same family. Theoretically, such performance should lie between the current method and Full-RL, and it would also allow a clearer comparison of the stylistic and performance differences between the "prefixes" obtained by this method and those obtained in the paper.
3. Has Prefix-RL shown gains on OOD tasks? For example, can we observe that the adapter model produces more guiding responses in tasks other than mathematical reasoning?
4. In L460–L461 it is mentioned that "cross-family configurations lead to performance degradation." Is there any data that can visually show the extent of these performance drops? Also, note that open-source models versus closed-source models would also count as "cross-family."

---

> ### Author Response · Authors · 2025-11-27
>
> We thank the reviewer for their thoughtful review and positive assessment of the work. Here we respond to weaknesses and questions raised by the reviewer.
>
> ---
>
> ### **Weakness 1: Same‑family requirement / closed‑source models (Q4)**
>
> We agree and now state this limitation clearly in the Discussion. Prefix‑RL works best when adapter and target share a family, because the prefix encodes family‑specific conventions (reasoning style, formatting, LaTeX habits, etc.). In preliminary cross‑family experiments (e.g., Qwen‑style adapter with a Llama target), we observed noticeable drops, so in the paper, we frame cross‑family adapters as future work rather than a solved problem.
>
> Importantly, this does **not** preclude closed‑source targets: Prefix‑RL only needs inference access. In many practical settings, there exists a small open‑weights model that is trained to mimic the provider’s larger model (or at least share a similar style), which can serve as the adapter.
>
> ---
>
> ### **Weakness 2: Only math tasks / applicability to other RLVR domains (Q3)**
>
> Our goal is an efficient way to run **RLVR**, and math is a clean testbed where rewards are deterministic and easy to verify. We now say this more explicitly: extending Prefix‑RL to general RLHF is out of scope.
>
> That said, we broadened the evaluation beyond the MATH train distribution:
>
> * two college‑level physics benchmarks, OCW Courses and UGPhysics (Table 7, Fig. 7).
>
> On all of these, Prefix‑RL improves pass@1 over the base model (typically by 3–7 points), showing gains on out‑of‑distribution math/physics tasks, while we still restrict claims to verifiable‑reward domains.
>
> ---
>
> ### **Q1: Why does Prefix‑Clustering behave anomalously on Qwen?**
>
> We have added a clearer explanation with examples. Qualitatively:
>
> * For **Llama**, high‑accuracy generations almost always begin with a generic step‑by‑step scaffold such as “Step 1: …”, regardless of the specific problem. A single fixed 16‑token prefix capturing this style therefore transfers well across questions.
> * For **Qwen**, good solutions usually start by restating *problem‑specific* structure (“We are given a triangle ABC…”, “To compute the integral of …”). The optimal prefix therefore depends strongly on the input; forcing one fixed 16‑token prefix can conflict with the question and hurt accuracy.
>
> This explains why a fixed, input‑independent prefix helps Llama but is brittle for Qwen; Prefix‑RL, which learns input‑conditional prefixes, does not suffer from this issue.
>
> ---
>
> ### **Q2: Using the target model directly as the adapter**
>
> There are two related cases:
>
> 1. **Same‑size adapter experiment (already in Sec. 2).** We set g to have the *same architecture and initialization* as the reference Llama‑1B model but allow it to emit only the first k tokens, with the reference completing the rest. This recovers a large fraction of full RL’s gains while changing only the prefix, supporting the “strategy selection” interpretation.
> 2. **Adapter \= large target.** If we literally take the full target (e.g., Qwen‑7B) as the adapter and update it with RL, then gradients must flow through all target weights; training FLOPs again scale with the target size, and the method no longer works in inference‑only settings. This collapses back to standard full‑model RL (possibly with a prefix restriction), so we focus on smaller adapters where Prefix‑RL’s compute and accessibility advantages apply.
>
> ---
>
> Thanks again for your constructive feedback. If these clarifications and new experiments address your concerns, we would hope you might consider increasing your overall score, and we would be happy to clarify anything that remains unclear.

---

### Official Review · Reviewer_hLfX · 2025-11-07

**Soundness:** 3
**Presentation:** 2
**Contribution:** 2
**Rating:** 4
**Confidence:** 4

**Summary:**

This paper introduces a methodology called _Prefix-RL_ to algorithmically identify ways for an LLM to start its response given a user input so that it is more likely to correctly answer math questions. The work uses this methodology to fine-tune both LLama and Qwen models on the mathematical reasoning benchmarks MATH, AIME, AMC23, and Minerva. The results show similar gains to direct RL finetuning.

**Strengths:**

* The paper's primary strength is the ~3000x reduction in FLOPs and the 4x reduction in GPU requirements during training. This approach to RL fine-tuning is much more accessible to research labs than tuning the full model.

* The core idea is based on an interesting insight and is also practical to implement. The insight that prefixes index into parts of the training data that are useful for answering certain math questions is potentially a fruitful idea for inspiring more works.

* This method is demonstrated on FP8-quantized models, which, as noted in Section 3.2, was previously difficult to do. It seems to make progress on the performance gap between quantized Llama-8B and its full-precision counterpart, which is impressive and useful.

**Weaknesses:**

*Weaknesses*
* The work is posed as a form of parameter-efficient RL, but only compares against a standard RL baselines. A more fair comparison would consider other techniques for parameter-efficient RL such as LoRA [Hu et al., 2022] (or QLoRA [Dettmers et al., 2023] for quantized models) or Adapters [Houlsby et al., 2019]. It would also be nice to compare against prefix-tuning, as mentioned in the related work, given that it can be directly trained on the same labels generated for RL as a supervised signal.

* It is unclear why the baseline method of prefix clustering was selected with k=16, whereas the experimental method was tested at k=32 and k=64. I believe this is a bit of an unfair comparison, as it is possible that k=16 is simply not enough tokens to meaningfully index into the parts of the LLMs training data that are “good” for solving math questions, which is the core insight of this work. It would be better to have a comparison that has equal numbers of k values across all approaches.

* Additionally, selecting k doesn’t seem to be very clear-cut. In Table 1, it appears that having k=32 seems to work well for some models/benchmarks (e.g., the Qwen-72B model or the Minerva Benchmark), whereas k=64 works better for others. It seems difficult to know ahead of time what value should be selected for k to achieve good performance. This mitigates some of the benefits of efficiency, because implementing this approach now requires an engineer to search over the k-values that work the best.

* The paper appears to report results from a single training run for each experiment. The training curves (e.g., in Figure 4) seem to be quite noisy, with high variance between steps. Selecting the "best checkpoint" from a single, noisy run is not super robust to initial conditions. The paper should report the mean and standard deviation over multiple runs (e.g., 3-5) with different random seeds to establish statistical significance, as is the standard in the RL literature.

* This approach seems to rely on having some objective way of calculating “correctness” of an answer. It is not clear how well this does under different kinds of label noise that is common in RLHF. Having a “correct answer” is a large assumption for problems that LLMs are typically used for, such as creative writing, open-ended dialogue, or exploratory information retrieval.

*Minor Edits*
* The section on Prefix clustering is a bit confusing. Is it one prefix for all evaluation examples, or is it the nearest cluster’s prefix? It seems like the former, but a more reasonable baseline would be the latter.
* It is not immediately clear why setting g_theta to the same size and architecture of the reference model implies that “improvement from RL upweights existing strategies” as claimed in Sec 2 under the subheading “Prefix-RL”. This could be better elaborated on.
* In Section 3.1, the authors state the MATH training split has 7,500 examples. This dataset is appears to have 12,500 examples. Your subsequent filtered number of 8,888 examples also suggests the starting number was larger than 7.5k. Please double-check and clarify this dataset statistic.

References:
Hu, Edward J., Yelong Shen, Phillip Wallis, Zeyuan Allen-Zhu, Yuanzhi Li, Shean Wang, Lu Wang, and Weizhu Chen. "Lora: Low-rank adaptation of large language models." ICLR 1, no. 2 (2022): 3.

Dettmers, Tim, Artidoro Pagnoni, Ari Holtzman, and Luke Zettlemoyer. "Qlora: Efficient finetuning of quantized llms." Advances in neural information processing systems 36 (2023): 10088-10115.

Houlsby, Neil, Andrei Giurgiu, Stanislaw Jastrzebski, Bruna Morrone, Quentin De Laroussilhe, Andrea Gesmundo, Mona Attariyan, and Sylvain Gelly. "Parameter-efficient transfer learning for NLP." In International conference on machine learning, pp. 2790-2799. PMLR, 2019.

**Questions:**

The questions below correspond to the number of bullet points of the weaknesses.

* 1.1: How does this technique compare to other techniques for parameter-efficient RL such as LoRA [Hu et al., 2022] (or QLoRA [Dettmers et al., 2023] for quantized models) or Adapters [Houlsby et al., 2019]?
* 1.2: What are the benefits of this approach of using RL with automatically calculated labels vs. supervised fine-tuning with automatic labels?

* 2.1: How does prefix clustering at k=32 and k=64 compare to the proposed approach?

* 3.1: how can one determine a k-value for their problem?
* 3.2: what is the worst-case number of evaluations to make for k?

* 4.1: how consistent are these results across different training runs?

* 5.1: how sensitive is this approach to noise in labels?
* 5.2: how applicable is this approach to other problems that are common uses of LLMs?

---

> ### Author Response · Authors · 2025-11-27
>
> We thank the reviewer for their thoughtful review and positive assessment of the work. Here we respond to weaknesses and questions raised by the reviewer.
>
> ---
>
> ### 1\. Comparison to parameter-efficient RL (LoRA / QLoRA / adapters / Prefix-Tuning)
>
> **1.1 LoRA / adapters.** We added *explicit parameter-efficient baselines* on Qwen-2.5-7B (Table 1):
>
> * **LoRA-RL** (rank-64 on Qwen-7B, same RLVR pipeline as Prefix-RL) improves MATH-500 only from 67.4→70.2 (+2.8) and even **hurts** AMC23 from 40.3→36.2 (-4.1), while Prefix-RL (k=32) reaches 74.4 on MATH-500 and 50.0 on AMC23 (+7.0 and \+9.7).
> * **Prefix-Tuning (SFT)** with 32 virtual tokens on the same auto-labeled data results in a collapse of performance (e.g., MATH-500: 67.4→45.4).
>
> Conceptually, LoRA/QLoRA/adapters still backprop through the full target, so training FLOPs scale with target size (N\_t). Prefix-RL only updates a separate adapter of size (N\_a); the large model is inference-only. As derived in Table 2 and App. B, training cost scales as (C\_{\\text{train}} N\_a R k) instead of (C\_{\\text{train}} N\_t R T), giving a reduction of roughly (N\_t T / (N\_a k)) (≈3000× for Qwen-1.5B→72B).
>
> **1.2 RL vs supervised finetuning with auto labels.** Both Prefix-RL and LoRA-RL use the same verifiable 0/1 reward (`math_verify`). RLVR optimizes pass@1 directly and can explore multiple completions per prompt. The supervised Prefix-Tuning baseline, trained on the same auto-generated solutions, overfits and degrades accuracy, whereas Prefix-RL consistently improves it. We clarify this distinction in the text.
>
> ---
>
> ### 2\. Prefix-Clustering and the choice of (k)
>
> Prefix-Clustering is now explicitly framed as a *diagnostic*, not a tuned competitor. For each model, we cluster candidate prefixes on MATH-train and select a **single, input-independent** 16-token prefix used for all test problems. Longer *fixed* prefixes (32–64 tokens) proved brittle in preliminary runs, especially for Qwen, where a global 32–64-token prefix often conflicts with the question, so we keep PC at (k=16) and state this in Sec. 3.2. Prefix-RL, in contrast, learns input-conditional prefixes and is evaluated for multiple (k).
>
> ---
>
> ### 3\. How to choose (k) and what is the cost of sweeping it?
>
> To address Q3.1–3.2, we added a **full (k)-sweep** for Qwen-7B with (k ∈ {1,4,8,16,32,64}) (Table 8, Fig. 9):
>
> * On MATH-500, *all* (k) values improve over the 67.4% base, clustering around 72–74%.
> * On AIME, AMC23, and Minerva, the very short prefix (k=1) can be slightly worse, but every (k ≥ 4) yields consistent gains (e.g., up to \+7.5 on AMC23 and \+4.8 on Minerva).
>
> We also evaluate the *same* Qwen-7B Prefix-RL checkpoints on two physics benchmarks (OCW Courses, UGPhysics; Table 7, Fig. 7). For all (k) except 64, the prefixes improve over zero-shot, with the strongest gains again for moderate (k ≈ 8–32).
>
> A simple rule of thumb is therefore: choose (k) from a coarse set such as ({8,16,32,64}); in practice, (k=32) works well across models and domains, and any (k ≥ 4) tends to be robust. Even if a practitioner trains 3–5 Prefix-RL runs for different (k), the *total* training FLOPs remain orders of magnitude smaller than a single full-model RL run, since compute scales with (N\_a k) rather than (N\_t T).
>
> ---
>
> ### 4\. Robustness across random seeds
>
> For our main configuration (Qwen-1.5B→Qwen-7B, (k ∈ {32,64})) we now report **four seeds** (Table 6, Fig. 6). Every seed improves over the base on all four benchmarks, and the effect sizes (e.g., \+6.0 to \+10.6 on AMC23) are 2–25× larger than the corresponding standard deviations. Due to compute limits we cannot replicate all 70B experiments with many seeds, but this setup covers the core Qwen-7B results.
>
> ---
>
> ### 5\. Verifiable rewards, label noise, and scope
>
> Our work is explicitly scoped to **RL with Verifiable Rewards (RLVR)**, where the reward is a deterministic 0/1 checker. Prefix-RL and standard RLVR share the same reward, so Prefix-RL is not more sensitive to label noise than existing RLVR pipelines. Beyond MATH-500, we now report results on AIME 1983–2024, AIME-2025, AMC23, Minerva, OlympiadBench (Tables 1, 5\) and on OCW Courses and UGPhysics (Table 7), where Prefix-RL consistently gives 3–7 point gains without catastrophic failures. We clearly mark open-ended RLHF tasks (dialogue, creative writing) as out of scope and list them as future work.
>
> ---
>
> We also fixed the minor issues you flagged (Prefix-Clustering description, “same-size adapter” explanation, and MATH dataset statistics).
>
> Thank you again for your constructive criticism. If these clarifications and new experiments address your concerns, we would be grateful if you could consider increasing your score. We are happy to clarify anything that remains unclear.

---

### Meta-Review · Area_Chair_e6gy · 2026-01-15

**Summary:**

According to the reviews, the decision-relevant concerns can be split into the following parts:

1. Strength and novelty. Reviewers agree the paper provides a clean diagnostic method and  propose Prefix-RL that can can recover a large fraction of full sequence RLVR gains. The main concerns was whether the claim were appropriately scoped.
2. Comparative baselines and fairness of comparisons. Several reviewers asked for a clearer comparison to full RL and fair comparison to parameter-efficient methods.
3. Robustness and hyperparameter sensitivity. The reviewers mentioned that the RL runs can be noisy, and requested multi-seed runs' results.

**Reviewer Concerns:**

Addressed by rebuttal and revision:
1. Comparison with parameter efficient baselines(LoRA/prefix-tuning)
2. Compute and flops measurement.
3. Robustness of the method over the seed, and the k selection.

Partially addressed:
1. Directly compare with the full-RL. The rebuttal cannot provide a true full-RL baseline at that scale due to budget.
2. Cross-family degradation quantification. The rebuttal does not provide a systematic comparison with cross-family methods.

Those remaining gaps are just about adding more proof, not messing with the main contribution.

**Reviewer Scores:**

- Reviewer Yk1Z (score 8) Already positive.
- Reviewer y1zy (score 4) Their main weaknesses were same-family limitation and limited domain coverage.
- Reviewer hLfX (score 4) The rebuttal addressed each point raised by reviewers, I would expect the reviewer will increate the score to 6 if they engaged into the discussion.
- Reviewer UfLZ (score 2) flagged as fully AI-generated.

---

### Decision · Program_Chairs · 2026-01-26

Accept (Poster)